# Confounding Factors Influencing the Kinetics and Magnitude of Serological Response Following Administration of BNT162b2

**DOI:** 10.3390/microorganisms9061340

**Published:** 2021-06-21

**Authors:** Jean-Louis Bayart, Laure Morimont, Mélanie Closset, Grégoire Wieërs, Tatiana Roy, Vincent Gerin, Marc Elsen, Christine Eucher, Sandrine Van Eeckhoudt, Nathalie Ausselet, Clara David, François Mullier, Jean-Michel Dogné, Julien Favresse, Jonathan Douxfils

**Affiliations:** 1Department of Laboratory Medicine, Clinique Saint-Pierre, 1340 Ottignies-Louvain-la-Neuve, Belgium; jean-louis.bayart@cspo.be (J.-L.B.); tatiana.roy@cspo.be (T.R.); vincent.gerin@cspo.be (V.G.); 2Namur Thrombosis and Hemostasis Center, Department of Pharmacy, Namur Research Institute for Life Sciences, University of Namur, 5000 Namur, Belgium; laure.morimont@unamur.be (L.M.); jean-michel.dogne@unamur.be (J.-M.D.); julien.favresse@unamur.be (J.F.); 3QUALIblood S.A., 5000 Namur, Belgium; clara.david@qualiblood.eu; 4Department of Laboratory Medicine, CHU-UCL Namur, Université Catholique de Louvain, 5530 Yvoir, Belgium; melanie.closset@uclouvain.be (M.C.); francois.mullier@uclouvain.be (F.M.); 5Department of Internal Medicine, Clinique Saint-Pierre, 1340 Ottignies-Louvain-la-Neuve, Belgium; gregoire.wieers@cspo.be; 6Department of Laboratory Medicine, Clinique Saint-Luc Bouge, 5004 Namur, Belgium; marc.elsen@slbo.be (M.E.); christine.eucher@slbo.be (C.E.); 7Department of Internal Medicine, Clinique Saint-Luc Bouge, 5004 Namur, Belgium; sandrine.vaneeckhoudt@slbo.be; 8Department of Internal Medicine, CHU-UCL Namur, Université Catholique de Louvain, 5530 Yvoir, Belgium; nathalie.ausselet@uclouvain.be

**Keywords:** SARS-CoV-2, vaccine, BNT162b2, antibody, serology, kinetic, age, gender, BMI, blood-group

## Abstract

Background: Little is known about potential confounding factors influencing the humoral response in individuals having received the BNT162b2 vaccine. Methods: Blood samples from 231 subjects were collected before and 14, 28, and 42 days following coronavirus disease 2019 (COVID-19) vaccination with BNT162b2. Anti-spike receptor-binding-domain protein (anti-Spike/RBD) immunoglobulin G (IgG) antibodies were measured at each time-point. Impact of age, sex, childbearing age status, hormonal therapy, blood group, body mass index and past-history of severe acute respiratory syndrome coronavirus 2 (SARS-CoV-2) infection were assessed by multivariable analyses. Results and Conclusions: In naïve subjects, the level of anti-Spike/RBD antibodies gradually increased following administration of the first dose to reach the maximal response at day 28 and then plateauing at day 42. In vaccinated subjects with previous SARS-CoV-2 infection, the plateau was reached sooner (i.e., at day 14). In the naïve population, age had a significant negative impact on anti-Spike/RBD titers at days 14 and 28 while lower levels were observed for males at day 42, when corrected for other confounding factors. Body mass index (BMI) as well as B and AB blood groups had a significant impact in various subgroups on the early response at day 14 but no longer after. No significant confounding factors were highlighted in the previously infected group.

## 1. Introduction

Vaccines against severe acute respiratory syndrome coronavirus 2 (SARS-CoV-2) are today the main hope for curbing the spread of infection worldwide. Among the several types of SARS-CoV-2 vaccines that have been developed, mRNA-based vaccines were the first to be approved by the European Medicines Agency (EMA) and the Food and Drug Administration (FDA) [1]. It has been found that the BNT162b2 vaccine (Comirnaty^®^; Pfizer-BioNTech; Puurs, Belgium and Mainz, Germany) conferred a protection of 95% against coronavirus disease 2019 (COVID-19) in a large and multinational clinical trial [2]. While the efficacy and safety data obtained from mass vaccination campaign are very encouraging, data concerning the humoral response following the administration of the two-dose regimen of the BNT162b2 vaccine are only emerging. More specifically, there is little information regarding possible confounding factors that may lead to variability in vaccine-induced immunogenicity.

In this study, we comprehensively characterized the early kinetics and magnitude of immunoglobulin G (IgG) antibody response against the SARS-CoV-2 receptor binding domain of the spike protein (Spike/RBD) in a cohort of 231 subjects. We also assessed whether the age, sex, ABO blood group, childbearing age status, hormonal therapy, body mass index (BMI) and previous SARS-CoV-2 infection were likely to influence the immune response.

## 2. Materials and Methods

### 2.1. Study Population

Two hundred and thirty-one volunteers from three medical centers in Belgium were enrolled in an ongoing prospective and interventional clinical trial (CRO-VAX-HCP study; EudraCT registration number: 2020-006149-21) [3]. The primary objective of this study was to assess the humoral response in a population of healthcare professionals having received the BNT162b2 mRNA COVID-19 vaccine. The demographic characteristics of the population are shown in Table 1.

Among them, 74% (*n* = 170) were females (mean age = 42.6 years; range: 23 to 66 years) and 26% (*n* = 61) were males (mean age = 42.8 years; range: 23 to 64 years). Seventy-three volunteers (31.6%) had a previous positive molecular diagnostic of SARS-CoV-2 infection (*n* = 65; mean time since reverse transcription polymerase chain reaction (RT-PCR) = 99 days) and/or a positive serological diagnostic at baseline evaluation (*n* = 8). Participants received the first vaccine dose from 18 January to 17 February 2021. The second vaccine dose was systematically administered 21 days after the first dose. Samples were collected within two days (i.e., defined as day 0) and after 14 (+2), 28 (+3) and 42 (+4) days following the first dose of BNT162b2. Demographic data were collected at baseline and included sex, age, ABO blood group, childbearing age status, female hormonal therapy and BMI.

All participants provided detailed informed consent prior to collection of data and specimen.

### 2.2. Analytical Procedures

Anti-spike receptor-binding-domain protein (anti-Spike/RBD) IgG antibodies (Architect^®^ SARS-CoV-2 IgG II Quant, Abbott, Wavre, Belgium) were measured at baseline and at 14, 28 and 42 days following the first dose administration. The positivity cut-off provided by the manufacturer (i.e., >50 arbitrary unit (AU)/mL) was used. Samples higher than the upper limit of linearity (40,000 AU/mL) were systematically diluted and retested. Total antibodies against the SARS-CoV-2 nucleocapsid (NCP) (Elecsys^®^ Anti-SARS-CoV-2 NCP qualitative ECLIA, Roche Diagnostics, Machelen, Belgium) were also measured to attest about a past-infectious episode with an optimized cutoff index (COI) of 0.165 as positive threshold [4].

### 2.3. Statistical Analysis

Multiple comparison between anti-Spike/RBD values at each time point were performed by Kruskal–Wallis followed by Dunn’s test. Stratifications were done according to the previous COVID-19 status, the sex, and the age (≤ or >45 years). Correlations between continuous variables and the log_10_-transformed anti-Spike/RBD values was done using the Spearman correlation coefficient. Direct comparison for dichotomous variables were assessed using the Mann–Whitney non-parametric test.

Multiple linear regression adjusted for COVID-19 pre-infectious status, sex, ABO blood group, hormonal status and therapy for female (dichotomous variables), age and BMI (continuous variables) were used to analyze the clinical determinants of the log_10_-transformed anti-Spike/RBD values outcome in the population at days 14, 28 and 42. Sub analyses have been performed using multiple stratifications based on pre-COVID-19 infection and sex. Multicollinearity was assessed to verify how each independent variable can be predicted from the other variables. R^2^ values for collinearity greater than 0.75 suspect that multicollinearity is present in the model.

*p* value < 0.05 was used as a significance level. Data analysis was performed using GraphPad Prism^®^ software (version 9.1.0, San Diego, CA, USA).

## 3. Results

### 3.1. Global Kinetics

In previously uninfected and seronegative individuals, 96.1% (148/154) showed a seroconversion 14 days after the first dose. Following the second dose, all subjects showed values above the positive threshold. At day 28, previously uninfected subjects showed a median value of 14,989 AU/mL (interquartile range (IQR): 7238 to 24,947 AU/mL), a 40.5-fold increase (*p* < 0.0001) compared to the serological response observed at day 14 (370 AU/mL; IQR: 181 to 831 AU/mL). At the individual level, volunteers reached their highest antibody titers at day 28 or 42. However, no significant change was observed between day 28 and day 42 (*p* > 0.999) (Table 2 and Figure 1). None of these uninfected volunteers were positive to anti-NCP antibodies at baseline and anti-NCP titers remained unchanged until day 42 (data not shown).

In individuals with a history of SARS-CoV-2 infection or positive anti-NCP at baseline, 94.5% (69/73) also presented with anti-Spike/RBD antibodies at baseline collection (299 AU/mL; IQR 174 to 540 AU/mL). Following vaccination, a significant (*p* < 0.0001) 78.8-fold increase was observed at day 14 (23,515 AU/mL; IQR: 15,793 to 35,088 AU/mL). Following the administration of the second dose, no significant increases were objectified at days 28 and 42, compared to day 14 (25,256 AU/mL; IQR: 16,182 to 34,731 AU/mL and 25,508 AU/mL; IQR: 16,802 to 40,657 AU/mL, respectively; *p* > 0.999) (Table 2 and Figure 1). Anti-NCP titers remained unchanged until day 42 (data not shown).

### 3.2. Influence of Previous Severe Acute Respiratory Syndrome Coronavirus 2 (SARS-CoV-2) Infection

The multivariable analysis revealed that previous SARS-CoV-2 status is an important confounder of the serological response at day 14 (β coefficient = 1.683; *p* < 0.0001), 28 (β coefficient = 0.0696; *p* < 0.0001) and 42 (β coefficient = 0.209; *p* < 0.0001) (Appendix A).

Compared to naïve individuals, anti-Spike/RBD titers of previously infected volunteers were statistically higher at day 14 (*p* < 0.0001). Higher medians were also observed at day 28 (25,256 AU/mL vs. 14,989 AU/mL) and day 42 (25,508 AU/mL vs. 15,591 AU/mL) but the differences were not statistically significant (*p* = 0.055 and *p* = 0.152, respectively) (Table 2).

### 3.3. Influence of Age and Sex

Multivariable analysis highlighted a significant decreased response with age in the naïve group at day 14 (β coefficient = −0.0147; *p* < 0.0001), which was less pronounced at day 28 (β coefficient = −0.008; *p* = 0.0381) and no more significant at day 42 (β coefficient = −0.004; *p* = 0.0731). This observation was also observed by means of Spearman correlations, both for women (r = −0.391, *p* < 0.0001) and men (r = −0.501, *p* = 0.0006) at day 14. These associations were attenuated following the second dose administration and remained only slightly statistically significant for women at day 28 (r = −0.204, *p* = 0.042) and for men at day 42 (r = −0.336, *p* = 0.042) (Figure 2).

The Table 2 reports anti-Spike/RBD median values when separating the groups for either sex and age (≤ or >45 years). We observed statistically significant differences between females ≤ or >45 years (718 vs. 264 AU/mL, *p* < 0.0001) and between males ≤ or >45 years (459 vs. 170 AU/mL, *p* < 0.0001). Among the same sex group, differences between the ≤45 and >45 age groups were no longer observed following administration of the second dose (Figure 2 and Table 2).

In volunteers previously infected with SARS-CoV-2, multivariable analysis did not reveal any impact of age on the magnitude of the antibody response (Appendix A). This lack of correlation was also observed by simple linear regression analysis (Figure 2).

Multivariable analysis did not reveal any statistical differences between men and women at days 14 and 28, when analyzing the full cohort or when stratifying by previous COVID-19 infection status. However, at day 42 men from the SARS-CoV-2 naïve group showed a significantly lower antibody response (β coefficient = −0.129; *p* = 0.0394) than women. No differences in sex were observed in volunteers previously infected with SARS-CoV-2 (Appendix A).

### 3.4. Influence of ABO Blood Group

At day 14, a negative impact on the serological response of blood group AB in the whole cohort was observed (β coefficient = −0.3152; *p* = 0.045) both in the female sex group (β coefficient = −0.476; *p* = 0.017) and in the childbearing age group of this latter population (β coefficient = −0.538; *p* = 0.0032). In the previously infected group, patients with B blood group also presented lower antibody titers at day 14 (β coefficient = −1.086; *p* = 0.001). These differences were no longer observed after 28 or 42 days (Appendix A).

### 3.5. Other Potential Confounding Factors

The multivariable analysis did not find any association between the serological response and the BMI in the two main subgroups of the study (i.e., COVID-19 positive or naïve). However, in women of childbearing age (independent of past SARS-CoV-2 status), multivariable analysis showed a positive correlation between their BMI and the serological response at day 14 (β coefficient = 0.0257; *p* = 0.0094) but this correlation was not significant at later time-points. When looking at the childbearing age group, we could not see a difference in the serological response between women taking hormonal contraceptive agents or not. The same was observed with hormonal replacement therapy in the menopausal group (Appendix A).

## 4. Discussion

Our results confirm that people with a previous SARS-CoV-2 infection develop a more rapid and more important serological response than naïve subjects. After day 14, an additional boosting effect of the second dose is not observed in this population. These results support some of the public health strategies currently being pursued. Indeed, to reach rapid control of the current pandemic, some countries adopted the strategy of postponing or omitting the second dose in this specific subgroup [5,6]. Whether a single dose could maintain comparable long-term immunity to the conventional scheme would need specific randomized or long-term observational studies.

At day 28, the whole cohort had seroconverted. In volunteers without a history of past SARS-CoV-2 infection, the second dose led to a 40.5-fold increase in anti-Spike/RBD antibodies compared to levels measured 14 days after the first dose. No further increase was observed at day 42. Median values observed in this group were systematically lower compared to previously infected people and the multivariable analysis confirmed that the previous SARS-CoV-2 status is the most important confounding factor, at each time-point investigated in the present study.

It is now well known that the symptoms and mortality of COVID-19 may vary according to several factors including ethnicity, sex, age, obesity and some comorbidities like deprivation, diabetes, or severe asthma [7,8,9,10]. The humoral adaptive immunity is a key factor to prevent viral cell penetration and current vaccine candidates have demonstrated robust humoral responses. A plethora of studies have already investigated factors influencing the humoral vaccine response and these include intrinsic host factors (such as age, sex, genetics and comorbidities), extrinsic factors (such as nutrition, smoking, alcohol consumption) and previous exposition to the infectious agent concerned [6,11,12,13,14]. Nevertheless, due to the rapid development and approval of current COVID-19 vaccines used, there are still limited data concerning potential confounding factors in COVID-19 vaccine recipients.

Several studies reported that females develop a more robust immune response compared to males. In 1967, Butterworth et al. reported that women produce higher levels of circulating immunoglobulins IgG and IgM than men, which was subsequently confirmed by multiple studies [15]. The biological reasons behind this could explain the observed female protection from COVID-19 fatal outcome [14]. On the other hand, several recent studies reported sex-specific differences in the kinetic evolution of anti-SARS-CoV-2 antibody titers following SARS-CoV-2 infection. These data pointed out higher anti-Spike (IgG and IgA) and higher neutralizing antibodies in men, possibly explaining the higher risk of adverse COVID-19 outcome in this latter population through a stronger inflammatory response [16,17,18]. However, some of these studies lack strength because they do not analyze confounding factors in a multivariable setting.

Regarding the immune response to vaccination, recent data from Kontopoulou et al. did not show differences in anti-Spike/RBD titers between men and women 14 days after BNT162b2 administration [19]. Kamal et al. reached the same conclusion 21 days after the first dose [20]. Accordingly, we did not observe significant differences at days 14 and 28. Nevertheless, in the present study, multivariable analysis identified men in the SARS-CoV-2 naïve group as lower responders compared to women at day 42. Terpos et al. also observed higher titers of neutralizing antibodies in women 22 and 50 days following first dose of the BNT612b2 vaccine. Of note, these observations concerned only octogenarians and the authors did not use a multivariable model adjusting the impact of sex for the effect of other variables, as we did [21]. More follow-up studies are, therefore, needed to analyze if this difference between genders is confirmed or not, and eventually if it increases over time.

The concept of immunosenescence, a state characterized by the weakening of the immunological response related to age, is well-known and has been described for several existing vaccines [22,23,24]. Moreover, decreased antibody production related to age was recently highlighted in several studies which measured anti-SARS-CoV-2 antibody levels at a single time-point following a unique dose of BNT162b2 [19,20,25]. These data conflict with conclusions issued from the largest clinical trial which suggested similar efficacy among different age groups receiving twice administration of the latter vaccine [2]. Indeed, the multivariable analysis identified age as a factor negatively impacting the serological response to the vaccine at days 14 and 28 in the SARS-CoV-2 naïve group. This negative trend which stronger at day 14, has been further illustrated by the Spearman correlation analysis. However, this trend was much less pronounced after the second dose and the impact of age was no more statistically significant at day 42. These observations are consistent with recent results obtained by Jalkanen et al. who observed that IgG anti-Spike and neutralizing antibodies decreased significantly in the older age group (55–65 years) compared to younger age groups (20–44 years) but neutralizing antibody levels were similar between the two groups three weeks after the second dose [26]. Furthermore, this negative correlation between age and the magnitude of the antibody response was not observed in our volunteers with a history of previous SARS-CoV-2 infection. Taken together, these observations support the fact that repeated antigen (directly inoculated or generated through cell mechanism) stimulation could lead to a normalization of the differences initially observed following one dose of BNT612b2. This further corroborates the homogeneous clinical efficacy data observed across the whole age range in real-life settings [2].

We also investigated whether other intrinsic confounding factors could influence the humoral immune response. Several studies reported relationship between the A blood group or obesity and COVID-19 susceptibility [27,28,29,30]. On the other hand, obesity is frequently pointed-out as a risk factor for vaccine non-responsiveness and a recent research suggest that people with blood group B have relatively higher neutralizing antibody titers following SARS-CoV-2 infection [31,32]. Interestingly, AB blood group was identified in the full cohort, in the female group as well as in the childbearing age group as a negative predictor of vaccine response at day 14. Blood group B was also negatively correlated with anti-Spike/RBD concentrations in the previous SARS-CoV-2 group, at the same time-point. These observations, due to the limited size of the subgroups of interest, would require broader investigations to investigate whether these blood groups do indeed show a lower initial immune response.

Finally, no further relationships were found with other potential confounding factors like hormone (contraception or hormonal replacement therapy) intake in this small cohort.

Our study has two main limitations. The first is the limited size of the cohort. Therefore, the impacts (or not) of several confounding factors highlighted here are exploratory and will require larger cohort studies to confirm the associations observed. However, although this study is of limited power, it generates interesting hypotheses that corroborate with data observed in natural SARS-CoV-2 infection or in the context of other vaccination programs. The second limitation is that we could not collect all potential confounding data for the whole population included in this study. However, the absence of multicollinearity in the confounding factors investigated give us confidence in the associations reported.

## 5. Conclusions

Among the multiple confounding factors investigated in this study, previous SARS-CoV-2 infection is the most powerful factor predicting the magnitude of the serological response. Age seems to play a role only in the SARS-CoV-2 naïve group and a negative impact was further attenuated following administration of the second dose. B and AB blood groups were negatively associated with early (day 14) antibody response in some specific subgroups. Finally, being a male was found to be a factor of reduced serological response in SARS-CoV-2 naïve patients, but only at the latest sampling time (day 42); this observation therefore requires longer follow-up studies to confirm this trend.

## Figures and Tables

**Figure 1 microorganisms-09-01340-f001:**
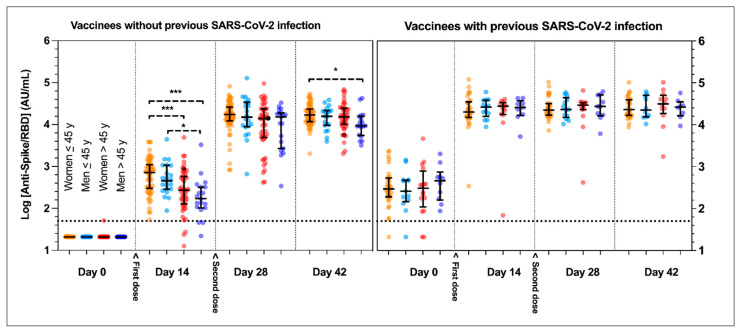
Kinetic response of anti-spike receptor-binding-domain protein (anti-Spike/RBD) antibodies in vaccinees without or with previous SARS-CoV-2 infection. Y-axes represent log_10_-transformed anti-Spike/RBD values. Horizontal dotted lines represent the positive threshold (= 50 AU/mL). The second dose was systematically administered at day 21. Groups were separated according to their previous coronavirus disease 2019 (COVID-19) status, sex and age (≤ or >45 years, a threshold chosen given the close median values observed in the two genders). Data are presented with median and interquartile range (IQR). *** *p* < 0.0001; * *p* < 0.05.

**Figure 2 microorganisms-09-01340-f002:**
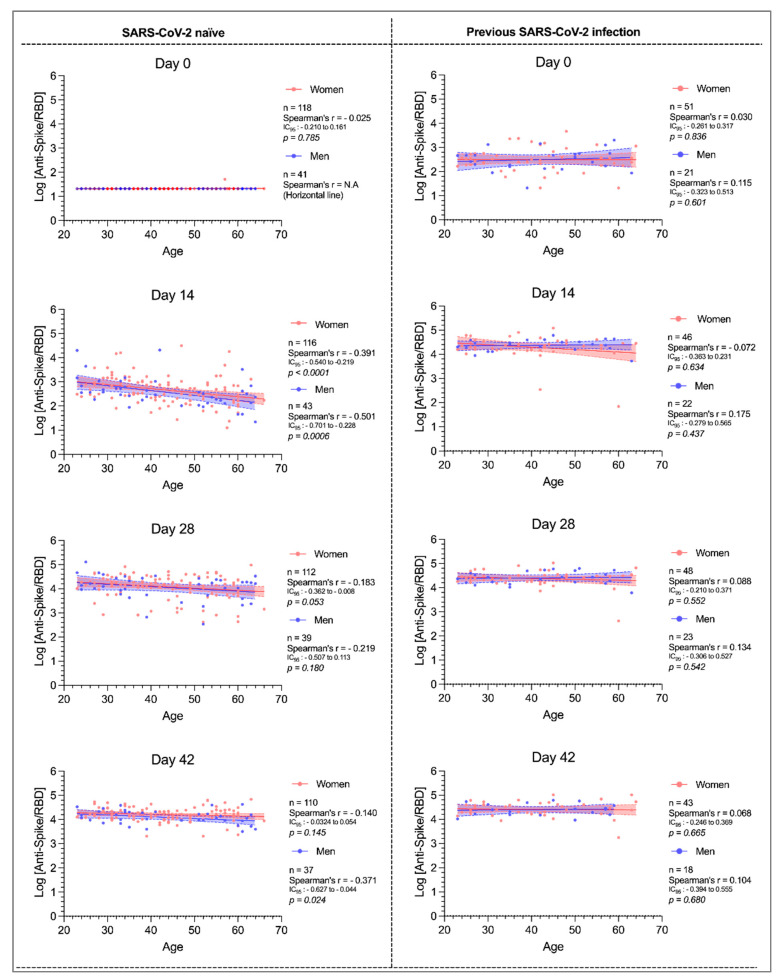
Simple linear regressions, with Spearman’s correlation coefficients, between log_10_-transformed anti-Spike/RBD values and age in previous or naïve SARS-CoV-2 infected people. The number of patients at the different timepoint may change due to loss of follow-up.

**Table 1 microorganisms-09-01340-t001:** Demographical characteristics of the participants.

Variables	Total Subject Number (*n* = 231)
**Age** (median, range)	43 (23–66)
≤45 years of age (*n*, %)	139 (40%)
*Previous SARS-CoV-2 infection (n,%)*	*47 (34%)*
>45 years of age (*n*, %)	92 (60%)
*Previous SARS-CoV-2 infection (n,%)*	*25 (27%)*
**Sex** (*n*, %)	
**Female**	169 (73%)
Previous SARS-CoV-2 infection (n,%)	51 (30%)
**Male**	62 (27%)
Previous SARS-CoV-2 infection (n,%)	21(34%)
**Previous SARS-CoV-2 infection** (*n*, %)	72 (31%)
Female (n, %)	51 (71%)
Male (n,%)	21 (29%)
≤45 years of age (n, %)	47 (65%)
>45 years of age (n, %)	25 (35%)
**BMI in kg/m^2^** (median, range) ^†^	23.7 (15.3–48.2)
**ABO blood group** (*n*, %)	
A	76 (33%)
B	9 (4%)
AB	18 (8%)
O	98 (42%)
Unknown	30 (13%)
**For female gender only**	
Childbearing age (*n*, %)	121 (72%)
*Hormonal contraception*	*78 (65%)*
*No hormonal contraception*	*38 (31%)*
*Unknown*	*5 (4%)*
Menopausal (*n*, %)	48 (28%)
*Hormonal replacement therapy*	*13 (27%)*
*No hormonal replacement therapy*	*30 (63%)*
*Unknown*	*5 (10%)*

† Body mass index (BMI) data were available only in 220 subjects.

**Table 2 microorganisms-09-01340-t002:** Median anti-Spike/RBD values at several time-points according to previous SARS-CoV-2 status, sex and age (≤ or > to 45 years).

**AU/mL**	**Women** **≤** **45 Years**	**Men** **≤** **45 Years**	**Women** **>45 Years**	**Men** **>45 Years**	***p*** **-Value**
Baseline	21 ^†^ (21–21)	21 ^†^ (21–21)	21 ^†^ (21–21)	21 ^†^ (21–21)	0.839
Day 14	718 (299–1111)	459 (284–1062)	271 (128–583)	170 (103–323)	<0.0001
Day 28	17343 (12,203–26,312)	14962 (8803–33,829)	13841 (4913–23,750)	15244 (2674–19,048)	0.0755
Day 42	16843 (11,768–23,427)	15784 (9530–21,602)	15372 (10,071–24,685)	9327 (5520–15,784)	0.0269
**AU/mL**	**Women** **≤** **45 Years**	**Men** **≤** **45 Years**	**Women** **>45 Years**	**Men** **>45 Years**	***p*** **-Value**
Baseline	293 (190–534)	260 (146–477)	304 (110–788)	462 (160–742)	0.849
Day 14	20,055 (15,047–35,055)	26,535 (15,722–38,628)	27,721 (17,683–33,703)	25,826 (16,659–37,683)	0.927
Day 28	22,270 (16,925–32,220)	23,236 (14,921–44,441)	29,584 (22,875–33,881)	27,402 (16,448–51,377)	0.905
Day 42	23,057 (16,857–39,774)	22,532 (15,244–50,159)	31,359 (18,646–50,229)	26,415 (16,388–34,968)	0.876

† 21 represents the lower limit of quantification of the test. Results below the value were reported as 21 for analytical purposes. Medians are presented with IQR. Statistical analyses were performed by Kruskal–Wallis test followed by Dunn’s multiple comparison test.

## Data Availability

The data presented in this study are available on request from the corresponding author. The data are not publicly available due to ethical and privacy reasons.

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
