# Peer review of "Confounding Factors Influencing the Kinetics and Magnitude of Serological Response Following Administration of BNT162b2"

_microorganisms, 2021, doi:10.3390/microorganisms9061340_

Round 1

Reviewer 1 Report

This is a reasonably designed study to find potential confounding factors influencing the humoral response elicited by the BNT162b2 vaccine using minimal clinical samples. Data presentation, explanation, and discussion in the manuscript are clear and pleasant. This manuscript is suitable for publication without further modification.

Author Response

We sincerely thank the reviewer for his/her encouraging comments.

Reviewer 2 Report

The authors assessed whether the confounding factors influencing anti-SARS-CoV-2 immune response after BNT162b2 vaccination in 231 subjects.

Although similar analyses have been already published, the topic of the manuscript is important and worth publishing. However, I do see the need for some clarifications and smaller corrections.

  1. Please indicate subject numbers of ≤ and > 45 years as well as numbers of un-infected persons (Both male and female) in Table 1.
  2. Please use ≤ throughout the manuscript including Figure 1 and Tables 1 & 2.
  3. Please add “2” after coronavirus.
  4. Are the unit of Y-axis same between Figure 1 and 2? It looks different to me.
  5. Please include Day0 data in Figure 2.
  6. Please confirm the number of men in Figure 2. There are 65 men at Day14.

Author Response

The authors assessed whether the confounding factors influencing anti-SARS-CoV-2 immune response after BNT162b2 vaccination in 231 subjects.

Although similar analyses have been already published, the topic of the manuscript is important and worth publishing. However, I do see the need for some clarifications and smaller corrections.

Response to the reviewer’s comment:

We thank the reviewer for his/her constructive review of the manuscript. We addressed all the comments and we corrected the manuscript accordingly.

  1. Please indicate subject numbers of ≤ and > 45 years as well as numbers of un-infected persons (Both male and female) in Table 1.

Response to the reviewer’s comment:

Thank you for this remark. We have reported this information in Table 1

  1. Please use ≤ throughout the manuscript including Figure 1 and Tables 1 & 2.

Response to the reviewer’s comment:

This has been adapted in the manuscript, table and figures. Thank you for this notice.

  1. Please add “2” after coronavirus.

Response to the reviewer’s comment:

This has been added.

  1. Are the unit of Y-axis same between Figure 1 and 2? It looks different to me.

Response to the reviewer’s comment:

This has been changed and harmonized.

  1. Please include Day0 data in Figure 2.

Response to the reviewer’s comment:

This has been changed accordingly.

  1. Please confirm the number of men in Figure 2. There are 65 men at Day14.

Response to the reviewer’s comment:

This has rechecked and is correct. We have some lost of follow-up. We have added a sentence in the legend to explain these differences.

Please see the attachment to have access to the manuscript including the changes that have been made according to your suggestions.
